# *In vitro* evidence to support amphotericin B and flucytosine combination therapy for talaromycosis

Heera Natesan Sambath[1]*, Shawin Vitsupakorn[1,2], Kaushik Sreerama Reddy[3], Lottie Brown[4,5], Thu Thi Mai Nguyen[1], Matthew Burke[1], Jialin Liu[1], Emily Evans[6], Charles Giamberardino[1], John Perfect[1], Ngo Thi Hoa[7,8,9], Thuy Le[1,7]*

1 Division of Infectious Diseases and International Health, Duke University School of Medicine, Durham, North Carolina, United States of America, 2 The Johns Hopkins University School of Medicine, Baltimore, Maryland, United States of America, 3 Division of Rheumatology and Clinical Immunology, University of Pittsburgh School of Medicine, Pittsburgh, Pennsylvania, United States of America, 4 Institute of Infection and Immunity, City St George's University of London, School of Health & Medical Sciences, London, United Kingdom, 5 Clinical Infection Unit, St George's Hospital, St George's University Hospitals NHS Foundation Trust, London, United Kingdom, 6 Emory University School of Medicine, Atlanta, Georgia, United States of America, 7 Tropical Medicine Research Center for Talaromycosis, Biomedical Research and Diagnostics Center, Pham Ngoc Thach University of Medicine, Ho Chi Minh City, Vietnam, 8 Oxford University Clinical Research Unit, Ho Chi Minh City, Vietnam, 9 Centre for Tropical Medicine and Global health, Nuffield Department of Medicine, University of Oxford, Oxford, United Kingdom

☯ These authors contributed equally to this work.
* Thuy.Le@duke.edu (TL) and Heera.Natesan.Sambath@duke.edu (HNS)

## Abstract

### Background

*Talaromyces marneffei* causes talaromycosis, a life-threatening fungal disease with limited treatment options. The standard treatment of amphotericin B (AmB) induction followed by itraconazole consolidation still results in 15% to 30% mortality. This study aimed to investigate the potential of AmB and flucytosine (5FC) combination therapy to enhance antifungal activity.

### Methods

The *in vitro* antifungal activity of AmB and 5FC alone and in combination against 60 *T. marneffei* clinical isolates was evaluated using a validated colorimetric antifungal susceptibility assay and the checkerboard method. The minimum inhibitory concentration (MIC) was defined as the lowest drug concentration inhibiting ≥ 95% fungal growth ($MIC_{95}$) for both AmB and 5FC. The combination effect between AmB and 5FC against *T. marneffei* was determined using fractional inhibitory concentration index. Combination effects were further tested using a time-kill assay.

**Data availability statement:** All relevant data are within the manuscript and its supporting information files.

**Funding:** This work was supported by grants from the National Institutes of Health to TL [R01AI143409, R21AI162367, R21AI159397, U01AI169358, P30AI064518]. LB is funded by the National Institute for Health Research (NIHR) and received fellowship grants from the Federation of European Microbiological Societies, Dowager Countess Eleanor Peel Trust, the British Society for Medical Mycology and the Wingate Foundation. EE receives support from a National Institutes of Health training grant (T32AI157855) and the Burrough's Wellcome Fund through the American Society of Tropical Medicine and Hygiene (Award ID: 0000089078). The funders had no role in the study design, data collection and analysis, decision to publish, or preparation of the manuscript.

**Competing interests:** I have read the journal's policy and the authors of this manuscript have the following competing interests: TL receives investigator-initiated research grant from Gilead Science to contribute to the ongoing phase III Liposomal Amphotericin B and Flucytosine Antifungal Strategy for Talaromycosis (LAmB-FAST) randomized clinical trial (clinicaltrials.gov NCT06525389). All other authors have declared that no conflicting interests exist.

## Results

The $MIC_{95}$ was $0.25 - 2$ µg/mL (geometric mean [GM] 0.68 µg/mL) for AmB, and $0.03 - 0.5$ µg/mL (GM 0.28 µg/mL) for 5FC. Full synergy was observed in 4 isolates (7%), and indifference was observed in the remaining 56 isolates (93%). The time-kill experiments revealed a concentration-dependent fungicidal activity of AmB, and concentration-independent fungistatic effect of 5FC. Synergy between AmB and 5FC was confirmed, showing greater than 2-$\log_{10}$ reduction in colony forming units when used in combination. No antagonism was observed.

## Conclusions

Our study demonstrated *in vitro* evidence of synergistic activity between AmB and 5FC against *T. marneffei*, providing the evidence to support *in vivo* and clinical trial testing of AmB and 5FC combination therapy, and dosing reduction strategies of 5FC.

## Author summary

Talaromycosis is an emerging, life-threatening, invasive fungal infection affecting people with weakened immune systems and is caused by the fungus *Talaromyces marneffei*, which is endemic to Southeast Asia. The mortality despite antifungal treatment is up to 30%, yet treatment options are limited to just two drugs: amphotericin B, which is associated with substantial drug toxicity, and itraconazole, which has poor absorption and limited effectiveness. In this study, we investigated whether combining amphotericin B with the existing antifungal drug flucytosine could improve antifungal activity against *Talaromyces marneffei*, compared to amphotericin B or flucytosine alone. These drugs inhibit fungal replication through different mechanisms, and their combination has been shown to improve tissue penetration and treatment outcomes in other fungal infections. Using checkerboard and time-kill assays on 60 clinical isolates obtained from talaromycosis patients, we found that the two-drug combination inhibited the growth of *Talaromyces marneffei* more effectively compared to the individual drugs. Our findings support the potential of combining amphotericin B and flucytosine as a more effective treatment strategy for talaromycosis and warrant further clinical evaluation. By strengthening the evidence for improved use of existing antifungal agents, our study contributes to advancing treatment options for this neglected fungal disease.

## Introduction

### The need to improve the treatment of talaromycosis

The thermally dimorphic fungus *Talaromyces marneffei* causes talaromycosis, a life-threatening invasive fungal disease endemic in Southeast Asia. Over just

three decades, talaromycosis has emerged from a rare disease to a leading opportunistic infection among people with advanced HIV and other immunocompromised conditions, with an estimated 25,000 cases annually [1]. Current treatment options for talaromycosis are limited to just two drugs: amphotericin B (AmB) for induction therapy for the first 2 weeks, followed by itraconazole consolidation and maintenance therapy for at least 12 months [2]. Despite appropriate therapy, mortality remains unacceptably high at 15% to 30% [3,4]. AmB is a large molecule with poor tissue penetration [5], which reduces its efficacy in treating deep-seated multi-organ infections. Treatment effect is further compromised by toxicity of AmB and poor pharmacokinetic profiles of itraconazole. Over 50% of patients on AmB experienced grade 3 or higher adverse events [2]. Itraconazole is better tolerated but has poor oral bioavailability (only 20% to 40%), non-linear pharmacokinetics, and a narrow therapeutic index [6]. The high mortality on treatment and limited treatment options prompt the inclusion of *T. marneffei* on the WHO Fungal Priority Pathogen List [7], calling for research on new therapeutics and treatment strategies for talaromycosis.

### Rationale for antifungal combination therapy

Some novel antifungal drugs have promising potential for talaromycosis but take a long time to be realized. A combination of existing therapy offers a promising strategy. Combination therapy using drugs from different antifungal classes has been used to optimize the treatment of invasive fungal diseases, including cryptococcosis, invasive aspergillosis and candidiasis [8–10]. Combining drugs with distinct and complementary mechanisms of action can enhance therapeutic efficacy by leveraging synergistic interactions, increasing drug penetration, accelerating fungal clearance, and ultimately improving patient outcomes. Enhanced antifungal activity could permit lower dosing, thereby reducing dose-related toxicity, and may help prevent the emergence of resistance associated with monotherapy [8].

### Rationale for amphotericin B and flucytosine (5FC) combination therapy for talaromycosis

AmB and 5FC act through different mechanisms which can potentiate one another to enhance fungal killing. *In vitro* studies of AmB and 5FC in *Cryptococcus* and *Candida* showed varied results ranging from synergy to indifference and, in the case of 5FC-resistant isolates, antagonism. Nevertheless, these findings translated into improved fungal clearance *in vivo* models and survival in clinical trials [11–15]. AmB and 5FC combination has been shown in several clinical trials of cryptococcal meningitis to enhance fungal clearance in the cerebrospinal fluid (CSF) and reduce mortality by 30% to 40% compared to AmB alone [16,17]. 5FC has excellent oral bioavailability, reaching a maximum serum concentration of 50 µg/mL within 2 hours [18], and good penetration into the deep tissues and CSF [18,19]. 5FC given at 25 mg/kg four times daily is generally safe and well-tolerated [16,17].

We and others have shown that 5FC has potent *in vitro* antifungal activity against *T. marneffei*. The minimum inhibitory concentrations (MICs) range between 0.015 and 1.0 µg/mL, which is 75-folds lower than serum concentrations achieved with standard dosing [20–22]. Given potent *in vitro* activity of 5FC against *T. marneffei,* robust mortality benefit of AmB and 5FC combination therapy in cryptococcal disease, and the favorable safety profile, we hypothesize that the AmB and 5FC combination may enhance the *in vitro* fungicidal activity against *T. marneffei* and aim to test this hypothesis with the goal to expand treatment options and improve outcomes of talaromycosis.

## Materials and methods

### Ethics statement

This study utilized *T. marneffei* clinical isolates obtained from participants of the multi-center Itraconazole *versus* Amphotericin for Penicilliosis (IVAP) randomized controlled trial, which was approved by all five study sites in Vietnam, and the Oxford Tropical Research Ethical Committee (OxTREC) in the UK, and the Vietnam Ministry of Health (MoH) (clinical trial registration: ISRCTN59144167, MoH approval: 3300/QD-BYT). All participants provided written informed consent

for specimens to be stored in the IVAP biobank at Duke University and used in this research (Duke IRB protocol: Pro00101915).

### *Talaromyces marneffei* isolates

For the current study, 60 *T. marneffei* clinical isolates were randomly selected from 336 isolates collected and whole-genome sequenced from the IVAP trial [2,3]. A sample size of 60 is considered more than sufficient to obtain a normal sampling distribution [23]. We selected 30 isolates from northern Vietnam and 30 from southern Vietnam to represent the two phylogenetically and geographically distinct clades [3]. The isolates were prepared as outlined in **S1 Appendix**.

### Antifungal susceptibility testing

Antifungal drugs were prepared as outlined in **S1 Appendix** and susceptibility testing was performed using a validated Clinical and Laboratory Standards Institute (CLSI)-based colorimetric assay [20], as outlined in **S1 Appendix**. Although a 95% inhibition ($MIC_{95}$) threshold was used as the MIC endpoint for AmB, and $MIC_{50}$ threshold was used for 5FC as per CLSI, $MIC_{95}$ was used for both AmB and 5FC to allow evaluation of combination effect by the checkerboard assay.

### AmB and 5FC combination effect in the checkerboard assay

**Fig 1** illustrates the checkerboard assay setup, which enables simultaneous determination of the MICs for each drug alone and in combination using a fungal inoculum of $1 - 5 \times 10^3$ colony forming units per mL (CFUs/mL). The combination effect of AmB and 5FC were characterized using the fractional inhibitory concentration index (FICI) [24], calculated according to **Equation A in S1 Appendix**. Classification of drug-drug interaction based on FICI values has varied across studies. Older studies classify FICI values between 0.5 and 1 as 'additive or partially synergy', and values between 1 and 4 as 'indifferent' [11,13]. Newer studies broadly define FICI values between 0.5 and 4 as 'indifference' [8,9]. As prior classification system may overestimate synergy, we used the more conservative and simple classification system to assess

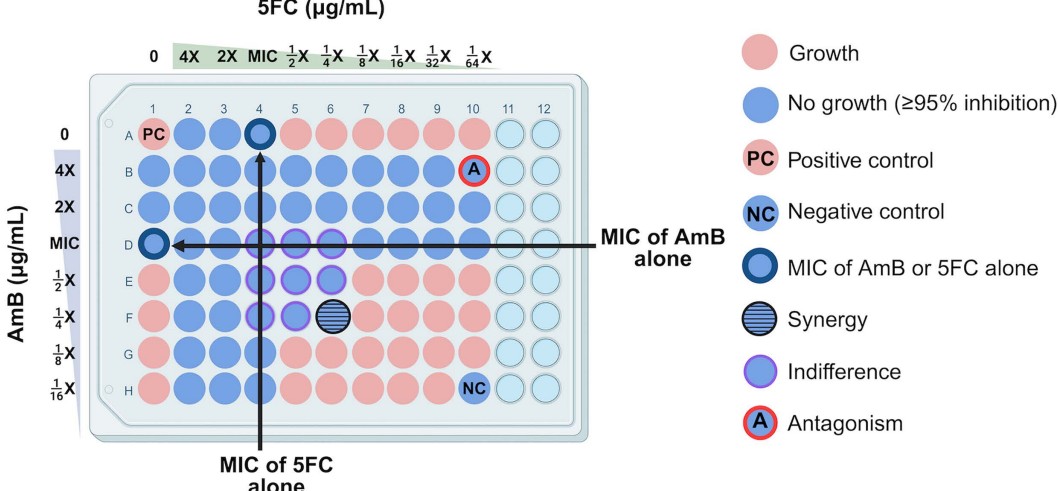

**Fig 1. Checkerboard assay method demonstrating the *in vitro* interaction between amphotericin B and flucytosine against *Talaromyces marneffei*.** The positive control well ("PC", position A1) contains no drug and allows for uninhibited growth. The negative control well ("NC", position H10) contained no inoculum. A well showing synergy (fractional inhibitory concentration index, FICI ≤ 0.5) is highlighted in blue with horizontal lines (position F6). Wells showing indifference (0.5 < FICI ≤ 4) are in blue with purple outline (positions D4 – D6, E4 – E6, F4, F5). A well showing antagonism (FICI > 4) is in blue with red outline ("A", position B10). Created in BioRender. Natesan Sambath, H. (2025) https://BioRender.com/h4r0z3t. Abbreviations: 5FC, flucytosine; AmB, amphotericin B; MIC, minimum inhibitory concentration.

interaction: [1] synergy (FICI ≤ 0.5), [2] indifference (0.5 < FICI ≤ 4), and [3] antagonism (FICI > 4). Synergy was defined as a minimum four-fold decrease in the MICs with AmB and 5FC combination compared to each drug alone.

### Validation of AmB and 5FC synergy by time kill assay

Time-kill assay was performed to evaluate the antifungal dynamics of AmB and 5FC combination over time. A *T. marneffei* isolate which showed evidence of synergy of AmB and 5FC combination in the checkerboard experiments was tested. Time-kill assay procedures are summarized in the **S1 Appendix**. Fungal growth was measured by CFUs/mL and converted to $\log_{10}$ CFU/mL to obtain normal distribution for analysis. A fungicidal effect of a drug was defined as a minimum $3\text{-}\log_{10}$ reduction in fungal burden from the starting inoculum, whereas a reduction of less than $3\text{-}\log_{10}$ is classified as fungistatic effect [25,26]. Synergy was defined as a reduction of at least $2\text{-}\log_{10}$ in fungal growth when the drugs are tested in combination compared to the most active agent alone [25].

### Quality control

Quality control procedures, including *Candida krusei* ATCC 6258 and the internal *T. marneffei* strain 11CN-20–091, are detailed in **S1 Appendix**.

### Statistical analysis

The mode and geometric mean (GM) of MICs, and the mean and standard deviation of the FICs and FICI for 60 isolates were calculated, along with 95% confidence interval (95% CI). The mean differences in MICs between groups – AmB alone, 5FC alone, *versus* their combination – were compared using paired student t-tests. MIC data follow a geometric sequence with a common ratio of 2 (i.e., 2-fold serial dilutions), therefore the data was $\log_2$ transformed to demonstrate fold-changes in **Fig 2**. For time-kill curves, we $\log_{10}$ transformed CFUs/mL and generated trajectories for each testing conditions over 120 hours. All statistical analyses were performed, and graphs were created using GraphPad Prism Version 8.4.0. (GraphPad software, Boston, MA, USA). Illustrations were created using BioRender (Toronto, Ontario, Canada).

## Results

### MICs of AmB and 5FC alone and in combination

**Table 1** summarizes the MICs for each drug, and the FICs and FICI of the 60 *T. marneffei* isolates. Individual values for each isolate are shown in **S1 Table**. The $MIC_{95}$ range for AmB was 0.25 – 2 µg/mL (GM 0.68 µg/mL). The $MIC_{95}$ and $MIC_{50}$ ranges for 5FC were 0.06 – 2 µg/mL (GM 0.28 µg/mL) and 0.03 – 0.5 µg/mL (GM 0.13 µg/mL), respectively. **Fig 2A and 2B** show the $MIC_{95}$ of AmB and 5FC when tested alone *versus* in combination. The MIC of AmB decreased three-fold on average when 5FC was added, and the MIC of 5FC decreased four-fold on average when AmB was added ($P < 0.0001$ for both groups by paired t-test).

### AmB and 5FC combination effect by checkerboard assay

**Fig 2C** shows the distribution of the FICI values and classification of combination effect for all isolates. Synergistic effects between AmB and 5FC were observed in four isolates (7%), all from patients in northern Vietnam. Indifferent effects were observed in the remaining 56 isolates (93%), and no antagonism was detected in any isolate.

### Concentration-dependent antifungal dynamics of AmB and 5FC against T. marneffei by time-kill assay

For the time-kill assay, we tested the *T. marneffei* isolate 11CN-21-012 which demonstrated synergy between AmB and 5FC in the checkerboard assay. **Fig 3A** demonstrates a concentration-dependent fungicidal effect of AmB against *T. marneffei* with a $3\text{-}\log_{10}$ reduction in CFUs/mL at concentrations of 0.5, 1, and 2 times the MIC compared to the starting

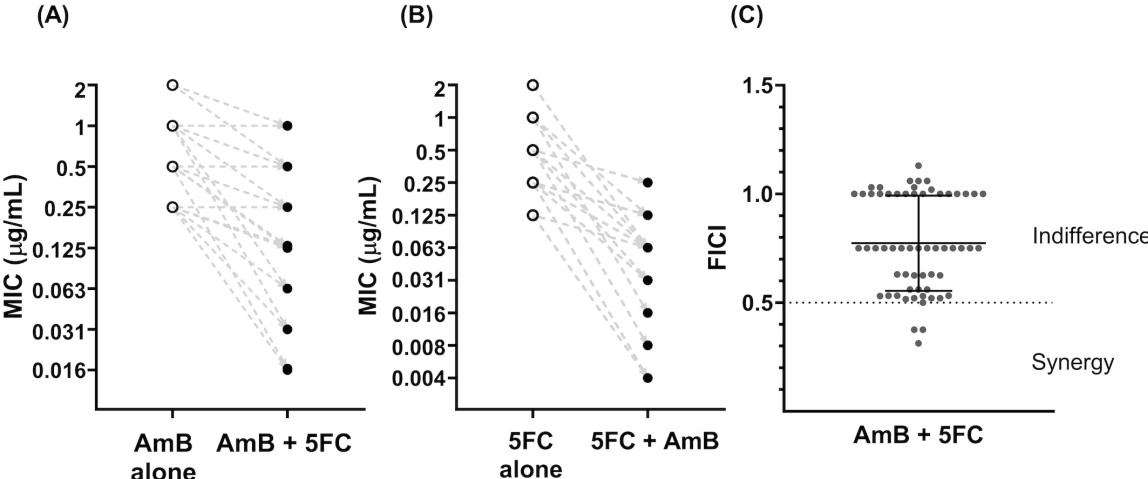

**Fig 2. Changes in the minimum inhibitory concentrations of amphotericin B and flucytosine when tested in combination *versus* alone, and the classification of combination effect observed for 60 *Talaromyces marneffei* clinical isolates. (A)** and **(B)** show the MICs of AmB and 5FC tested alone and in combination, with the dashed lines showing the directionality of changes between the MIC alone and in combination for each isolate. Absolute MIC data follows geometric sequence with a common ratio of 2, i.e., 2-fold serial dilution, therefore $\log_2$ MIC values were plotted to explain the fold reduction in MICs. **(A)** Minimum inhibitory concentrations (MIC) at 95% inhibition of amphotericin B (AmB) when tested alone *versus* in combination with flucytosine (5FC) against 60 *T. marneffei* clinical isolates. The MIC of AmB decreased three-fold on average when combined with 5FC (Paired t-test, $P < 0.0001$). **(B)** MIC of 5FC tested alone *versus* in combination with AmB against 60 *T. marneffei* isolates. The MIC of 5FC decreased four-fold on average when combined with AmB (Paired t-test, $P < 0.0001$). **(C)** Distribution of the fractional inhibitory concentration index (FICI) for AmB and 5FC combination in 60 *T. marneffei* isolates. Data are presented as individual FICI data points, with mean and standard deviation. Abbreviations: 5FC, flucytosine; AmB, amphotericin B; FICI, fractional inhibitory concentration index; MIC, minimum inhibitory concentration.

**Table 1. Summary of the minimum inhibitory concentrations and the combination effect between amphotericin B and flucytosine against 60 *Talaromyces marneffei* clinical isolates.**

| | MIC$_{95}$ (µg/mL) | | FIC$_{AmB}$ | MIC$_{95}$ (µg/mL) | | FIC$_{5FC}$ | FICI FIC$_{AmB}$+FIC$_{5FC}$ |
|---|---|---|---|---|---|---|---|
| | AmB | AmB+5FC | | 5FC | 5FC+AmB | | |
| ***Synergy** (FICI ≤ 0.5), n = 4 (7%)* | | | | | | | |
| Mean | 1.68 | 0.42 | 0.25 | 0.35 | 0.04 | 0.14 | 0.39 |
| | (95% CI: 1.20-2.36) | (95% CI: 0.30-0.59) | (0) | (95% CI: 0.15-0.85) | (95% CI: 0.02-0.09) | (0.08) | (0.08) |
| ***Indifference** (0.5 < FICI ≤ 4), n = 56 (93%)* | | | | | | | |
| Mean | 0.64 | 0.23 | 0.46 | 0.27 | 0.06 | 0.34 | 0.80 |
| | (95% CI: 0.55-0.75) | (95% CI: 0.18-0.31) | (0.23) | (95% CI: 0.22-0.34) | (95% CI: 0.04-0.08) | (0.25) | (0.20) |
| ***All isolates**, n = 60* | | | | | | | |
| Mean | 0.68 | 0.24 | 0.45 | 0.28 | 0.06 | 0.33 | 0.77 |
| | (95% CI: 0.58-0.80) | (95% CI: 0.19-0.32) | (0.23) | (95% CI: 0.22-0.34) | (95% CI: 0.04-0.08) | (0.25) | (0.22) |
| Mode | 1 | 0.5 | 0.5 | 0.25 | 0.06 | 0.5 | 0.75 |

The MICs of AmB and 5FC were defined as the lowest drug concentration that resulted in at least 95% inhibition of fungal growth. The geometric means with the 95% confidence intervals are reported for the MICs of AmB and 5FC, alone, and in combination. The arithmetic means with the standard deviations are reported for FICs of AmB and 5FC, and the FICI for 60 isolates.

Abbreviations: 5FC, flucytosine; 95% CI, 95% confidence interval; AmB, amphotericin B; FIC, fractional inhibitory concentration; FICI, fractional inhibitory concentration index; MIC, minimum inhibitory concentration.

inoculum. 5FC however demonstrated a concentration-independent antifungal effect, with less than 2-$\log_{10}$ reduction compared to the starting inoculum, even at concentrations as high as 20 times the MIC, and was thus considered fungistatic against *T. marneffei* (**Fig 3B**).

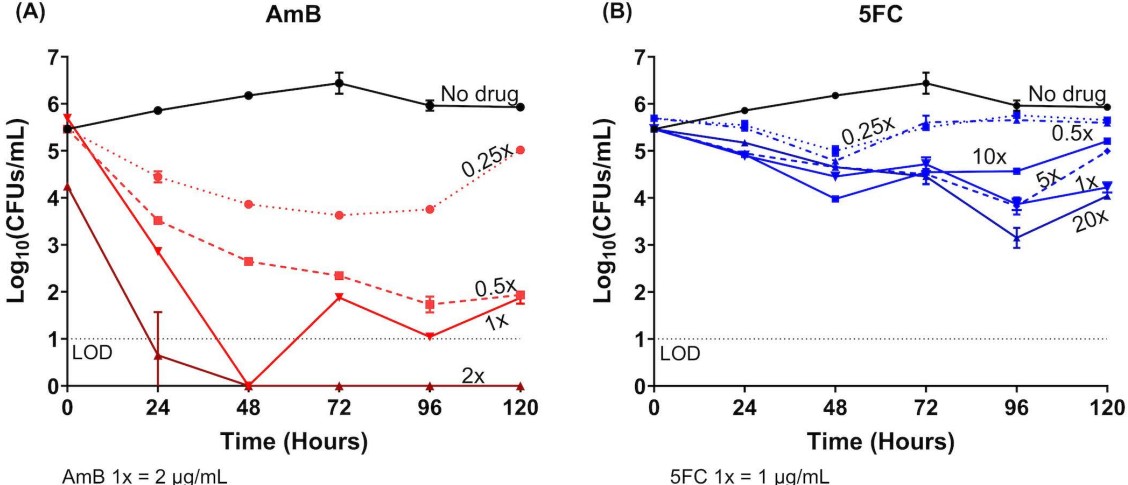

**Fig 3. Time-kill assay demonstrating the concentration-dependent fungicidal effect of amphotericin B and the concentration-independent fungistatic effect of flucytosine, tested alone against *Talaromyces marneffei* isolate 11CN-21-012. (A)** The fungicidal effects of AmB testing at concentrations 0.25 to 2 times the minimum inhibitory concentration (MIC) of AmB. **(B)** The fungistatic effects of 5FC testing at concentrations 0.25 to 20 times the MIC of 5FC. AmB showed fungicidal activity, resulting in a 3-log$_{10}$ reduction in colony forming units per milliliter (CFUs/mL) at concentrations of 0.5 or above the MIC by 96 hours. 5FC resulted in less than a 1 to 2-log$_{10}$ CFUs/mL reduction in the fungal count over 120 hours and was thus considered fungistatic against *T. marneffei.* Abbreviations: 5FC, flucytosine; AmB, amphotericin B; MIC, minimum inhibitory concentration; LOD, limit of detection; CFU, colony forming unit.

## Synergy between AmB and 5FC combination by time-kill assay

Because AmB exhibited such a robust fungicidal activity at its MIC (**Fig 3A**), and this concentration would have masked any potential synergistic effects of 5FC, we tested sub-MIC concentrations of AmB in combination with 5FC in order to allow a clear delineation of any additional antifungal effect of 5FC on AmB. Sub-MIC concentrations of AmB also mimic true drug concentrations achieved in the CSF due to poor tissue penetration of AmB [27].

The combinations of AmB (at 0.25 and 0.5 times the MIC) and a broad range of 5FC concentrations (1 – 20 times the MIC) resulted in a 2- to 3-log$_{10}$ reduction in CFUs by 96 hours compared to AmB alone. This is indicative of robust synergy between AmB and 5FC. Synergistic effect was evident even at the lowest concentrations of AmB and 5FC (0.25 times and 1 time the MIC, respectively) (**Fig 4**). Increasing the concentration of either drugs beyond these levels did not lead to further enhancement of antifungal activity. No evidence of antagonism was observed up to 120 hours.

## Discussion

### Summary of major findings

The current treatment of talaromycosis relies on only two drugs with suboptimal efficacy and safety profiles [2]. Antifungal combination therapy has shown to improve fungicidal activity and clinical outcomes for other serious fungal diseases, but has not been investigated for talaromycosis [9]. In this study, we performed a robust *in vitro* investigation of the antifungal effect of AmB and 5FC combination against *T. marneffei,* testing 60 *T. marneffei* clinical isolates representing two distinct phylogenetic and geographical clades in northern and southern Vietnam. We demonstrated full synergy (more than four-fold reduction of MICs bidirectionally) in 7% of isolates against *T. marneffei* using the checkerboard method. The remaining 93% of isolates showed a FICI between 0.52 and 1.13, which is classified by the more conservative classification system as indifference interaction. However, all 56 isolates demonstrated a three- to four-fold reduction in MICs with the AmB and 5FC combination, which would be classified as partial synergy by previous classification systems. We confirmed

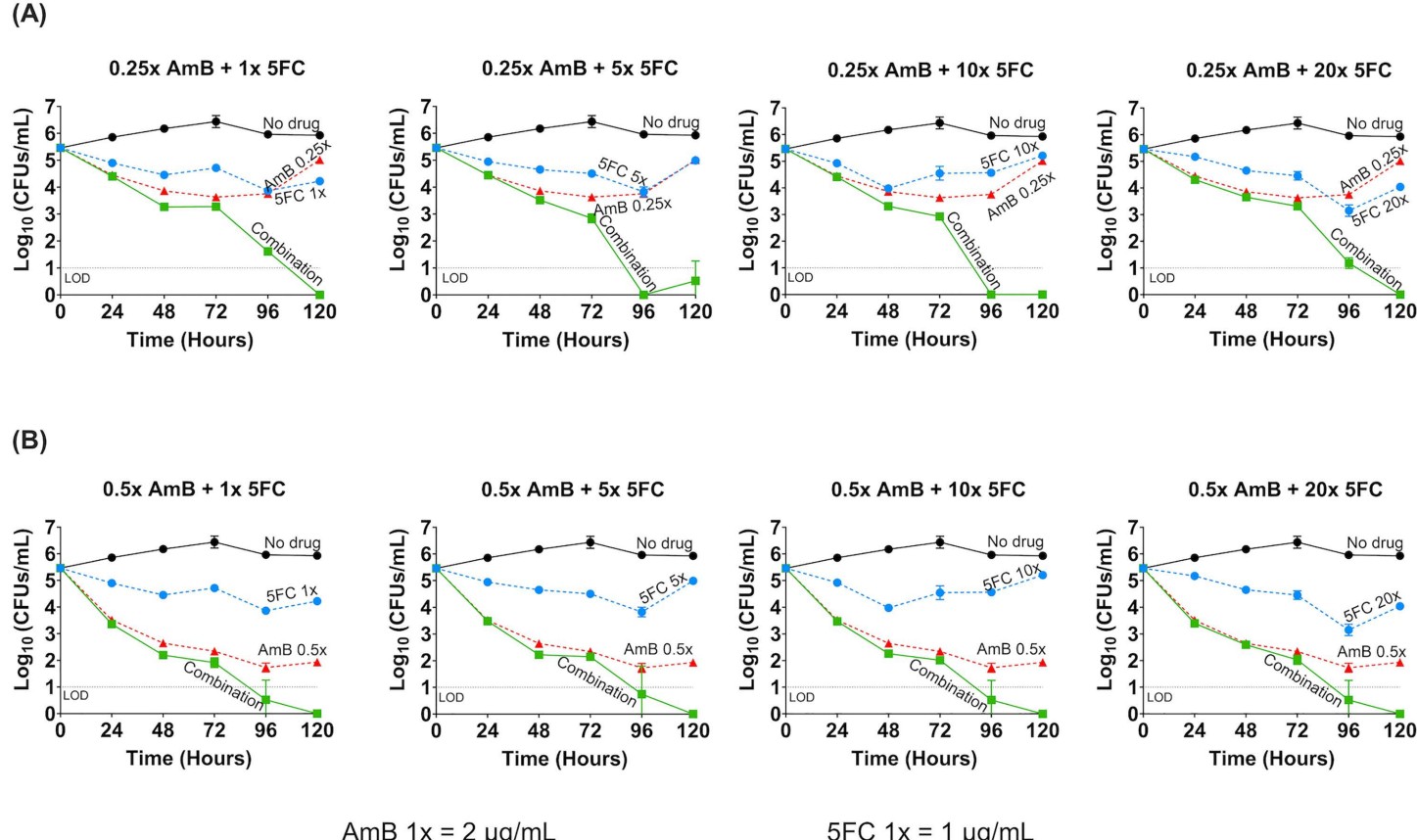

**(A)**

**(B)**

AmB 1x = 2 µg/mL          5FC 1x = 1 µg/mL

**Fig 4. Time-kill assay demonstrating the synergy of amphotericin B and flucytosine combination, tested at sub-MIC concentrations of amphotericin B in combination with various concentrations of flucytosine against *Talaromyces marneffei* isolate 11CN-21-012.** Amphotericin B (AmB, red) at 0.25 times the MIC **(A)** and 0.5 times the MIC **(B)** concentrations were tested in the time kill assay, as these sub-MIC concentrations were not rapidly fungicidal (see the results presented in **Fig 3**), thus permitting the evaluation of the combination. Both AmB concentrations were tested with flucytosine (5FC, blue) at 1, 5, 10, and 20 times the MIC. *T. marneffei* growth dynamics were measured by colony forming units (CFUs) over time at 0-, 24-, 48-, 72-, 96-, and 120-hours post-inoculation. The no-drug control group (black) was used to determine growth inhibition. At both AmB concentrations of 0.25 and 0.5 times the MIC, the AmB and 5FC combination (green) showed a ≥ 2-$\log_{10}$ CFUs/mL reduction when compared to AmB (the more active agent) or 5FC alone, demonstrating a synergistic effect between the combination. Antagonism was not seen, even at the highest concentration of 5FC over 120 hours. Abbreviations: 5FC, flucytosine; AmB, amphotericin B; MIC, minimum inhibitory concentration; LOD, limit of detection; CFU, colony forming unit.

full synergy with the time-kill experiment. No antagonism was observed. Together, these findings provide promising *in vitro* evidence that AmB and 5FC combination therapy may be beneficial against talaromycosis.

## Postulated mechanisms of synergy

We posit that the synergy between AmB and 5FC is due to their complementary mechanisms of action and permeability hypothesis. AmB extracts ergosterol from the fungal cell membrane, leading to leakage of intracellular ions and cell death [28]. This allows 5FC to permeate the cell wall and access the nucleus, where it converts to 5-fluorouracil and other metabolites that disrupt DNA and RNA synthesis [29]. AmB may also impair the fungal efflux mechanisms, thereby reducing the export of 5FC and resulting in higher intracellular concentration of 5FC [9]. Conversely, 5FC disrupts the pyrimidine biosynthesis, leading to structural defects in the fungal cell wall, thereby potentiating the fungicidal effect of AmB [29]. A

major pharmacokinetic limitation of AmB is its large molecular size with variable penetration in the lungs, liver, kidneys, and poor penetration in the brain and CSF [30,31]. The synergistic effect of 5FC on AmB may compensate for the pharmacokinetic limitation of AmB, and improve the treatment of deep-seated and multi-organ dissemination nature of talaromycosis. Enhanced clearance of deep-seated infection may also reduce the incidence of disease relapse, and incidence of immune reconstitution disease as the immune system of the host is restored, such as during antiretroviral therapy in patients with HIV [18].

### Our findings in the context of existing literature

The MIC ranges observed for AmB and 5FC tested alone were consistent with the literature and reflect the diversity of the clinical isolates. The four isolates showing full synergy also had higher MICs for AmB (1–2 µg/mL); however, not all isolates with higher AmB MICs exhibited full synergy. The remaining isolates exhibited indifference, consistent with previous *in vitro* studies on *Cryptococcus* and *Candida* spp*.,* which have reported a range of outcomes from synergy to indifference [26,32]. The evidence of *in vitro* synergy in *Cryptococcus* has translated into robust antifungal effect and mortality reduction in several randomized controlled trials, supporting AmB and 5FC combination therapy as the first-line induction regimen for cryptococcosis [16,33,34]. While *in vitro* synergy of AmB and 5FC combination against *Candida* has been inconsistent, excellent *in vivo* activity has been observed. Mice treated with AmB and 5FC combination had substantially higher survival rates than those treated with AmB or 5FC monotherapy (79.5% *vs* 15.9% and 13.7%, respectively) [35,36]. AmB and 5FC combination therapy is recommended for deep-seated *Candida* infections, such as *Candida* endocarditis and endophthalmitis [37]. *In vivo* studies and clinical trials are now needed to confirm the synergy of AmB and 5FC against *T. marneffei*, with the ultimate goal to reduce mortality from talaromycosis.

### Clade-specific difference in antifungal susceptibility

Interestingly, all four isolates showing full synergy were from northern Vietnam, raising the question of whether antifungal effect is clade-specific [3]. We have previously shown, in a genome wide association study of the 336 human *T. marneffei* isolates collected from the IVAP trial patients, that isolates from northern and southern Vietnam form two distinct geographical and phylogenetic clades [3]. Although we observed no significant differences in the MICs of AmB or itraconazole between the two clades nor clade-specific treatment responses, there were significant associations between the southern clade and biomarkers of disease severity, i.e., higher median baseline fungal load (2.5 *vs* 2.1 $\log_{10}$ CFU/mL, $P=0.024$, by Wilcoxon rank sum test) and absence of fever (34% *vs* 10%, $P=0.0002$, by Fisher's exact probability test) [3]. Additionally, genomic analysis showed reduced diversity in the southern clade, suggesting that it represents a more recently evolved lineage [3]. Clade-specific differences in azole resistance levels have been documented for other WHO fungal priority pathogens: *Candidozyma* (formerly *Candida*) *auris* and *Cryptococcus neoformans* [38–41]*,* suggesting that variation in antifungal combination activity is plausible. However, to our knowledge, no studies have systematically compared antifungal combination activity between genetic clades of yeasts, molds, or dimorphic fungi. Previous studies of antifungal combinations in these species either do not stratify outcomes by clades or lack direct comparisons between clades [39,42,43], and warrant further investigations.

### Have we been overdosing 5FC in the setting of antifungal combination therapy?

5FC toxicity, including myelosuppression and hepatotoxicity, is dose dependent [19]. The standard four times daily dosing of 5FC is impractical and can lead to adherence issues. Emerging evidence suggests that lower doses of 5FC maintain efficacy while reducing toxicity. The current recommended dose of 5FC (100mg/kg/day in 4 divided doses) rapidly achieves peak serum concentrations of 60 – 80 µg/mL [19], which is more than 75-fold the MICs of 5FC against *T. marneffei*. In the time-kill experiments, we showed that 5FC was fungistatic regardless of concentration, and that the synergistic effect of adding 5FC was not dose-dependent, suggesting that a higher 5FC dose or level is not more effective. In a

murine model of cryptococcal meningitis, the antifungal activity of 5FC at a 50mg/kg/day combined with AmB caused near maximal fungal killing [33], similar to the findings from a murine model of invasive candidiasis [44]. However, a recent non-comparative clinical trial in Uganda (n = 48) evaluated 5FC at a reduced dose (60mg/kg/day in 3 divided doses) for patients with cryptococcal meningitis and observed slower CSF fungal clearance rate than historical data for the standard 5FC dosing [45]. This is potentially an issue of 5FC penetration into the CSF, which need to be confirmed by pharmacokinetic data. Our data suggest that lower and/or less frequent dosing of 5FC may offer a more practical, safer, and equally effective alternative to the standard dosing of 5FC for talaromycosis.

## Strengths and limitations

Our study has several limitations. Firstly, our evaluation of combination effect in the 60 clinical isolates was based on the checkerboard method, which provides a "snapshot" of fungal growth inhibition at a fixed point in time. Disagreement between different methods of measuring drug activity (such as the checkerboard assays, E-test, and time-kill assay) has been reported [26]. While we investigated the temporal interaction between AmB and 5FC in a time-kill assay, our analysis was limited to a single isolate due to resource constraints, so the findings may not be generalizable across genetically diverse isolates. For efficiency, we recommend the checkerboard method as a screening tool for synergy and antagonism, followed by further assessment with time-kill assay and *in vivo* studies. Secondly, no consensus exists for calculating the FICI for agents with discordant endpoints (e.g., $MIC_{50}$ for 5FC *versus* $MIC_{95}$ for AmB). We chose $MIC_{95}$ threshold for 5FC in the evaluation of combination therapy to allow a compatible calculation of FICI in the checkerboard assay. Thirdly, only isolates from Vietnam were included in our study. Future studies should test isolates from a range of endemic regions to determine whether the fungicidal activity of AmB and 5FC is consistent across different geographical and genetic clades of *T. marneffei* beyond Vietnam. Finally, we recognize that *in vitro* interactions may not reflect real-world conditions, where variations in pharmacokinetics, pH, protein binding, host immune response, and the presence of other drugs and co-infections can impact clinical efficacy. *In vivo* studies are essential to understand the pharmacokinetic and pharmacodynamic properties of combination therapy. Despite these limitations, our study has notable strengths. We tested a large number of representative clinical isolates, undertook rigorous quality assurance measures, including the use of an internal control strain in all experiments and CFU counts to ensure inoculum consistency.

## Clinical implications

Antifungal combination therapy may improve treatment efficacy in deep-seated and disseminated talaromycosis, reducing the risk of relapse and mortality, and allowing for lower and less toxic dosage. Our study identifies AmB and 5FC as a promising synergistic combination and supports its testing in the ongoing Liposomal Amphotericin B and Flucytosine Antifungal Strategy for Talaromycosis (LAmB-FAST) clinical trial (NCT06525389). Our findings also pave the way for broader testing of antifungal combination therapies against talaromycosis and other invasive fungal infections.

## Supporting information

**S1 Appendix. Additional methods information.**
(DOCX)

**S1 Table. MIC, FIC, and FICI of 60 isolates.**
(DOCX)

**S1 Data. Time-kill assay.**
(XLSX)

## Acknowledgments

We thank the patients of the IVAP trial for donating samples for this research and healthcare staff in Vietnam who led recruitment.

## Author contributions

**Conceptualization:** Heera Natesan Sambath, Shawin Vitsupakorn, Kaushik Sreerama Reddy, Thu Thi Mai Nguyen, Thuy Le.

**Data curation:** Heera Natesan Sambath, Shawin Vitsupakorn, Thu Thi Mai Nguyen.

**Formal analysis:** Heera Natesan Sambath, Shawin Vitsupakorn, Thu Thi Mai Nguyen, Jialin Liu.

**Funding acquisition:** Thuy Le.

**Investigation:** Heera Natesan Sambath, Shawin Vitsupakorn, Kaushik Sreerama Reddy, Thu Thi Mai Nguyen, Matthew Burke.

**Methodology:** Heera Natesan Sambath, Shawin Vitsupakorn, Kaushik Sreerama Reddy, Thu Thi Mai Nguyen, Charles Giamberardino.

**Resources:** Thuy Le.

**Supervision:** Thuy Le.

**Validation:** Heera Natesan Sambath, Shawin Vitsupakorn, Kaushik Sreerama Reddy, Thu Thi Mai Nguyen, Charles Giamberardino.

**Visualization:** Heera Natesan Sambath, Shawin Vitsupakorn.

**Writing – original draft:** Heera Natesan Sambath, Shawin Vitsupakorn, Kaushik Sreerama Reddy.

**Writing – review & editing:** Heera Natesan Sambath, Shawin Vitsupakorn, Kaushik Sreerama Reddy, Lottie Brown, Thu Thi Mai Nguyen, Matthew Burke, Emily Evans, Charles Giamberardino, John Perfect, Hoa Thi Ngo, Thuy Le.

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
