## [Decision Letter · Decision Letter 0]

19 Nov 2025

*In Vitro*
Response to Reviewers
Revised Manuscript with Track Changes
Manuscript

Shaden Kamhawi

co-Editor-in-Chief

Paul Brindley

co-Editor-in-Chief

**Journal Requirements:**

1) Please upload all main figures as separate Figure files in .tif or .eps format. For more information about how to convert and format your figure files please see our guidelines: 

2) We have noticed that you have uploaded Supporting Information files, but you have not included a list of legends. Please add a full list of legends for your Supporting Information files after the references list.

**Reviewers' comments:**

**Key Review Criteria Required for Acceptance?**

**Methods**

-Are the objectives of the study clearly articulated with a clear testable hypothesis stated?

-Is the study design appropriate to address the stated objectives?

-Is the population clearly described and appropriate for the hypothesis being tested?

-Is the sample size sufficient to ensure adequate power to address the hypothesis being tested?

-Were correct statistical analysis used to support conclusions?

-Are there concerns about ethical or regulatory requirements being met?

Reviewer #1: (No Response)

**Results**

-Does the analysis presented match the analysis plan?

-Are the results clearly and completely presented?

-Are the figures (Tables, Images) of sufficient quality for clarity?

Reviewer #1: (No Response)

**Conclusions**

-Are the conclusions supported by the data presented?

-Are the limitations of analysis clearly described?

-Do the authors discuss how these data can be helpful to advance our understanding of the topic under study?

-Is public health relevance addressed?

Reviewer #1: (No Response)

**Editorial and Data Presentation Modifications?**

Reviewer #1: (No Response)

**Summary and General Comments**

Reviewer #1: 1.The text states: "Conversely, AmB is a large molecule that has variable penetration in the lungs, liver, kidneys, and a poor penetration in the brain and CSF (30,31). The disruption of pyrimidine biosynthesis by 5FC leads to structural defects in the fungal cell wall, facilitating membrane and tissue penetration of AmB and potentiating the fungicidal effect of AmB (29)."(Line 323-327)

The current phrasing, particularly the use of "Conversely... facilitating... tissue penetration of AmB", could be misinterpreted. The reason is:

The "tissue penetration of AmB" (e.g., in lungs, brain) is a pharmacokinetic concept, describing the drug's distribution in human organs and across barriers like the blood-brain barrier.

The "disruption of the fungal cell wall by 5FC" is a pharmacodynamic concept, describing the drug's effect on the pathogen's microstructure.

Linking them in this way might incorrectly suggest that the damage to the fungal cell wallby 5FC enhances the penetration of AmB through human tissues.

2.The paper notes that all four isolates showing full synergy were from northern Vietnam, raising the question of clade-specific treatment effects. Could the authors please discuss or speculate on the potential molecular mechanisms that might underlie this interesting observation? Furthermore, have similar geographically or phylogenetically clade-specific synergistic effects of antimicrobial combinations been reported in other important fungi, such as Candidaor Cryptococcusspecies?

PLOS authors have the option to publish the peer review history of their article (what does this mean? ). If published, this will include your full peer review and any attached files.

**Do you want your identity to be public for this peer review?** For information about this choice, including consent withdrawal, please see our Privacy Policy .

Reviewer #1: No

**Figure resubmission:**

**Reproducibility:** To enhance the reproducibility of your results, we recommend that authors of applicable studies deposit laboratory protocols in protocols.io, where a protocol can be assigned its own identifier (DOI) such that it can be cited independently in the future. Additionally, PLOS ONE offers an option to publish peer-reviewed clinical study protocols. Read more information on sharing protocols at https://plos.org/protocols?utm_medium=editorial-email&utm_source=authorletters&utm_campaign=protocols

---

## [Editor Report · Decision Letter 1]

19 Dec 2025

Dear Dr Natesan Sambath,

We are pleased to inform you that your manuscript '*In Vitro* Evidence to Support Amphotericin B and Flucytosine Combination Therapy for Talaromycosis' has been provisionally accepted for publication in PLOS Neglected Tropical Diseases.

Best regards,

Joshua Nosanchuk, MD

Section Editor

Shaden Kamhawi

co-Editor-in-Chief

Paul Brindley

co-Editor-in-Chief

---

## [Editor Report · Acceptance letter]

Dear Dr Natesan Sambath,

We are delighted to inform you that your manuscript, "*In Vitro* Evidence to Support Amphotericin B and Flucytosine Combination Therapy for Talaromycosis," has been formally accepted for publication in PLOS Neglected Tropical Diseases.

Best regards,

Shaden Kamhawi

co-Editor-in-Chief

Paul Brindley

co-Editor-in-Chief
